# CLUSTERING EMBEDDING TABLES, WITHOUT FIRST LEARNING THEM.

## ABSTRACT

To work with categorical features, machine learning systems employ embedding tables. These tables can become exceedingly large in modern recommendation systems, necessitating the development of new methods for fitting them in memory, even during training.

Some of the most successful methods for table compression are Product- and Residual Vector Quantization (Gray & Neuhoff, 1998). These methods replace table rows with references to k-means clustered "codewords." Unfortunately, this means they must first know the table before compressing it, so they can only save memory during inference, not training. Recent work has used hashing-based approaches to minimize memory usage during training, but the compression obtained is inferior to that obtained by "post-training" quantization.

We show that the best of both worlds may be obtained by combining techniques based on hashing and clustering. By first training a hashing-based "sketch", then clustering it, and then training the clustered quantization, our method achieves compression ratios close to those of post-training quantization with the training time memory reductions of hashing-based methods.

We show experimentally that our method provides better compression and/or accuracy that previous methods, and we prove that our method always converges to the optimal embedding table for least-squares training.

## 1 INTRODUCTION

Machine learning can model a variety of data types, including continuous, sparse, and sequential features. Categorical features are especially noteworthy since they necessitate an "embedding" of a (typically vast) vocabulary into a smaller vector space in order to facilitate further calculations. IDs of different types, such as user IDs, post IDs on social networks, video IDs, or IP addresses in recommendation systems, are examples of such features.

Natural Language Processing is another prominent use for embeddings (usually word embeddings such as Mikolov et al., 2013), however in NLP the vocabulary can be significantly reduced by considering "subwords" or "byte pair encodings". In recommendation systems like Matrix Factorization or DLRM (see fig. 2) it is typically not possible to factorize the vocabulary this way, and embedding tables end up very big, requiring hundreds of gigabytes of GPU memory (Naumov et al., 2019). This in effect forces models to be split across may GPUs which is expensive and creates a communication bottleneck during training and inference.

The traditional solution has been to hash the IDs down to a manageable universe size using the Hashing Trick (Weinberger et al., 2009), and accepting that unrelated IDs may wind up with the same representation. Too aggressive hashing naturally hurts the ability of the model to distinguish its inputs by mixing up unrelated concepts and reducing model accuracy.

Another option is to quantize the embedding tables. Typically, this entails rounding each individual parameter to 4 or 8 bits. Other quantization methods work in many dimensions at the same time, such as Product Quantization and Residual Vector Quantization. (See Gray & Neuhoff (1998) for a survey of quantization methods.) These multi-dimensional methods typically rely on *clustering* (like $k$-means) to find a set of representative "code words" to which each original ID is assigned. For example, vectors representing "red", "orange" and "blue" may be stored as simple "dark orange"

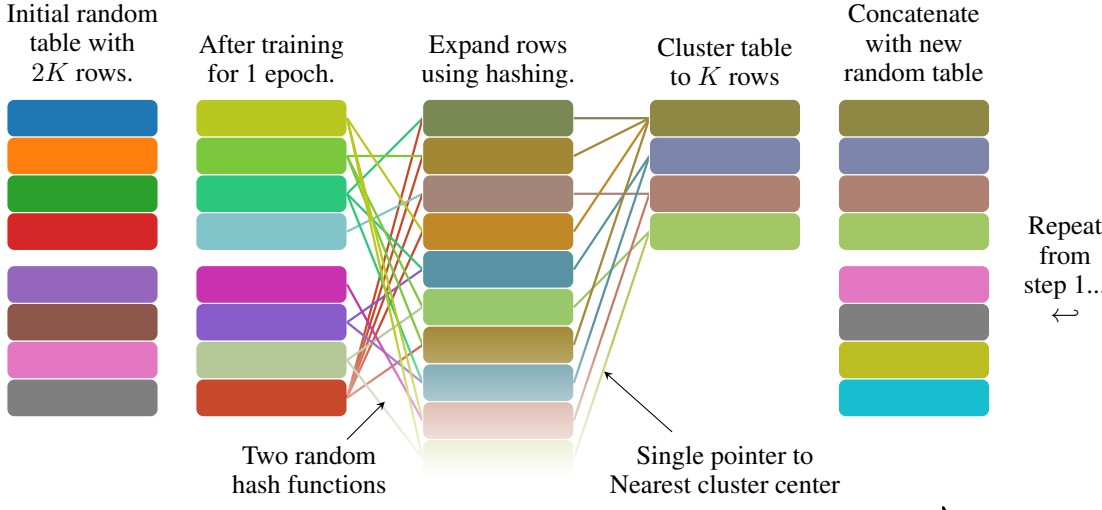

Figure 1: **Single iteration of Clustered QR.** Starting from a random embedding table, each ID is hashed to a vector in each of 2 small tables (left), and the value (shown in the middle) is taken to be the mean of the two vectors. After training for an epoch, the large (implicit) embedding table is (sub-sampled and) clustered. This leaves a new small table in which similar IDs are represented by the same vector. We can choose to combine the cluster centers with a new random table (and new hash function), after which the process can be repeated for an increasingly better understanding of which ID should be combined.

and "blue" with the two first concepts pointing to the same average embedding. See fig. 1 for an example. Even in the theoretical literature on optimal vector compression, such clustering plays a crucial role (Indyk & Wagner, 2022). All these quantization methods share one obvious drawback compared to hashing: the model is only quantized *after* training, thus memory utilization *during* training is unaffected. (Note: While it is common to do some "finetuning" of the model after, say, product quantization, the method remains primarily a "post-training" approach.)

Recent authors have considered more advanced ways to use hashing to overcome this problem: Tito Svenstrup et al. (2017); Shi et al. (2020); Desai et al. (2022); Yin et al. (2021); Kang et al. (2021). The common theme has been using multiple hash functions which allow features to take different representations with high probability, while still mapping into a small shared table of parameters. While these methods can work better than the hashing trick in some cases, they still fundamentally mix up completely unrelated concepts in a way that introduces large amounts of noise into the remaining machine learning model.

Clearly there is an essential difference between "post-training" compression methods like Product Quantization which *can utilize similarities between concepts* and "during training" techniques based on hashing, which are forced to randomly mix up concepts. This paper's key contribution is to bridge that gap: We present a novel compression approach we call "Clustered Compositional Embeddings" (or CQR for short) *that combines hashing and clustering while retaining the benefits of both methods.* By continuously interleaving clustering with training, we train recommendation models with accuracy matching post-training quantization, while using a fixed parameter count and computational cost throughout training, matching hashing based methods.

In spirit, our effort can be likened to methods like RigL (Evci et al., 2020), which discovers the wiring of a sparse neural network during training rather than pruning a dense network post training. Our work can also be seen as a form of "Online Product Quantization", though prior work like Xu et al. (2018) focused only on updating code words already assigned to concept. Our goal is more ambitious: We want to learn *which* concepts to group together without ever knowing the "true" embedding for the concepts.

*Why is this hard?* Imagine you are training your model and at some point decide to use the same vector for IDs $i$ and $j$. For the remaining duration of the training, you can never distinguish the two IDs again, and thus any decision you make is permanent. The more you cluster, the smaller your

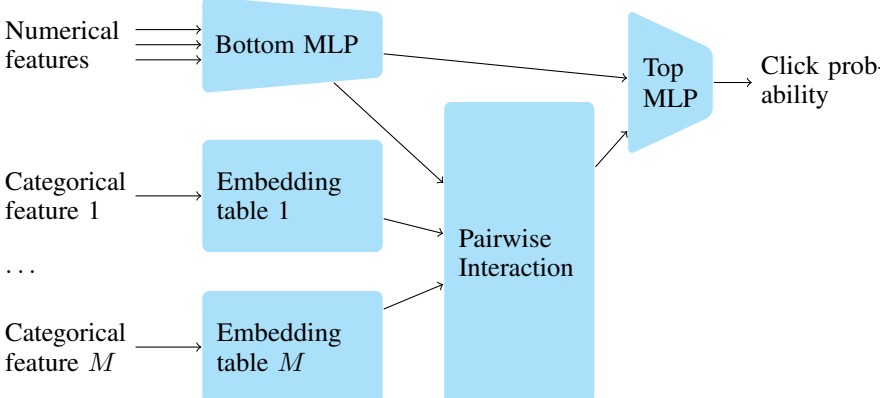

Figure 2: **Typical Recommendation System Architecture:** The DLRM model Naumov et al. (2019) embeds each categorical feature separately and combines the resulting vectors with pair-wise dot products. Other architectures use different interaction layers or a single embedding table for all categorical features, but the central role of the embedding table is universal.

table gets. But we are interested in keeping a constant number of parameters throughout training, while continuously improving the clustering.

In summary, our main contributions are:

- We show that it is possible to use clustering at training time, even though we don't have memory enough to learn or even store the concepts be clustered.
- Our method bridges the state of the art for quantizing "post-training" (clustering based methods) and during training (hashing methods).
- We show experimentally that our technique allows training the Deep Learning Recommendation System (DLRM) to baseline accuracy with less than 50% of the table parameters of the best previously suggested methods.
- We prove theoretically that a version of our method always succeeds in finding the optimal embedding table in the context of Least Squares learning.

## 2 BACKGROUND AND RELATED WORK

We show how most previous work on table compression can be seen in the theoretical framework of linear dimensionality reduction. This allows us to generalize many techniques and guide our intuition on how to choose the quality and number of hash functions in the system.

We omit standard common preprocessing tricks, such as weighting entities by frequency, using separate tables and precision for common vs uncommon elements, or completely pruning rare entities. We also don't cover the background of "post training" quantization, but refer to the survey, Gray & Neuhoff (1998). Finally, we keep things reasonably heuristic, but for a deep theoretical understanding of metric compression, we recommend Indyk & Wagner (2022).

### 2.1 EMBEDDING TABLES AS LINEAR MAPS

An embedding table is typically expressed as a tall skinny matrix $T \in \mathbb{R}^{d_1 \times d_2}$, where each ID $i \in [d_1]$ is mapped to the $i$-th row, $T[i]$. Alternatively, $i$ can be expressed as a one-hot row-vector $e_i \in \{0, 1\}^{d_1}$ in which case $T[i] = e_i T \in \mathbb{R}^{d_2}$.

Most previous work in the area of table compression is based on the idea of sketching: We introduce a (typically sparse) matrix $H \in \{0, 1\}^{d_1 \times k}$ and a dense matrix $M \in \mathbb{R}^{k \times d_2}$, where $k << d_1$, and take $T = HM$. In other words, to compute $T[i]$ we compute $(e_i H)M$. Since $H$ and $e_i$ are both sparse, this requires very little memory and takes only constant time. The vector $e_i H \in \mathbb{R}^k$ is called "the sketch of $i$" and $M$ is the "compressed embedding table" that is trained with gradient descent.

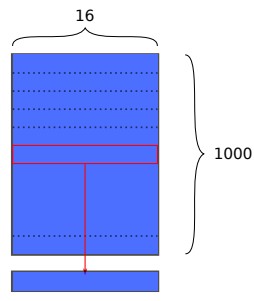
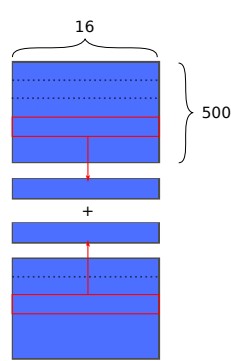

(a) **The Hashing Trick:** Each ID is hashed to one location in a table (here with 1000 rows) and it is assigned the embedding vector stored at the location. Many IDs are likely to share the same vector.

(b) **Hash Embeddings:** Each ID is hashed to two rows, one per each table, and its embedding vector is assigned to be the sum of those two vectors. Here, we use two separate tables unlike in Tito Svenstrup et al. (2017).

Figure 3: The Hashing Trick and Hash Embeddings shown side by side with an equal amount of parameters.

In this framework we can also express most other approaches to training-time table compression:

**The Hashing Trick** (Weinberger et al., 2009) is normally described by a hash function $h : [d_1] \to [k]$, such that $i$ is given the vector $M[h(i)]$, where $M$ is a table with just $k \ll d_1$ rows. Alternatively we can think of this trick as multiplying $e_i$ with a random matrix $H \in \{0, 1\}^{d_1 \times k}$ which has exactly one 1 in each row. Then the embedding of $i$ is $M[h(i)] = e_i H M$, where $HM \in \mathbb{R}^{d_1 \times d_2}$.

**Hash Embeddings** (Tito Svenstrup et al., 2017) map each ID $i \in V$ to the sum of a few table rows. For example, if $i$ is mapped to two rows, then its embedding vector is $v = M[h_1(i)] + M[h_2(i)]$. Using the notation of $H \in \{0, 1\}^{m \times n}$, one can check that this corresponds to each row having exactly two 1s. In the paper, the authors also consider weighted combinations, which simply means that the non-zero entries of $H$ can be some real numbers.

**Compositional embeddings** (or QR embeddings, Shi et al., 2020), define $h_1(i) = \lfloor i/p \rfloor$ and $h_2(i) = i \mod p$ for integer $p$, and then combines $T[h_1(i)]$ and $T[h_2(i)]$ in various ways. As mentioned by the authors, this choice is, however, not of great importance, and more general hash functions can also be used, which allows for more flexibility in the size and number of tables. Besides using sums, like Hash Embeddings, the authors also propose element-wise multiplication[1] and concatenation. Concatenation $[T[h_1(i)], T[h_2(i)]]$ can again be described with a matrix $H \in \{0, 1\}^{d_1 \times k}$ where each row has exactly one 1 in the top half of $H$ and one in the bottom half of $H$, as well as a block diagonal matrix $M$. While this restricts the variations in embedding matrices $T$ that are allowed, we usually compensate by picking a larger $m$, so the difference in entropy is not much different from Hash Embeddings, and the practical results are very similar as well.

**ROBE embeddings** (Desai et al., 2022) are essentially compositional embeddings with concatenation as described above, but adds some more flexibility in the indexing, including the ability for chunks to "wrap around" in the embedding table. In our experiments ROBE was near indistinguishable from QR-concat for large models, though it did give some measurable improvements for very small tables.

**Deep Hashing Embeddings** DHE (Kang et al., 2021) picks 1024 hash functions $h_1, \ldots, h_{1024} : [d_1] \to [-1, 1]$ and feed the vector $(h_1(i), \ldots, h_{1024}(i))$ into a multi-layer perceptron. While the idea of using an MLP to save memory at the cost of larger compute is novel and parts from the sketching framework, the first hashing step of DHE is just sketching with a

---

[1]While combining vectors with element-wise multiplication is not a linear operation, from personal communication with the authors, it is unfortunately hard to train such embeddings in practice. Hence we focus on the two linear variants.

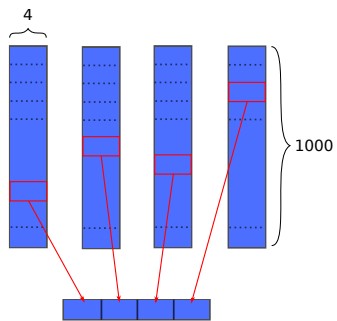

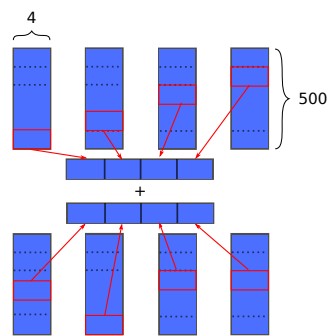

(a) **QR with concatenation:** In the hashing version of compositional embeddings (QR), each ID is hashed to a location in each of, say, 4 different tables. The four vectors stored there are concatenated into the final embedding vector. Given the large number of possible combinations (here $1000^4$), it is unlikely that two IDs get assigned the exact same embedding vector, even if they may share each part with some other IDs.

(b) **QR hybrid:** We can combine the sum hashing method of Tito Svenstrup et al. (2017) with the concatenation method of Shi et al. (2020). Each ID then gets assigned a vector that is the concatenation of smaller sums. In itself this method is not particularly interesting, but it is an essential step towards the Clustered QR method we describe in this paper.

Figure 4: Compositional Embeddings with concatenation and the Hybrid method which we will introduce later on.

> dense random matrix $H \in [-1,1]^{d_1 \times 1024}$. While this is less efficient than a sparse matrix, it can still be applied efficiently to sparse inputs, $e_i$, and stored in small amounts of memory.
>
> Unfortunately, in our experiments, DHE did not perform as well as the other methods, unless the MLP was taken to have just one layer, in which case it is just a linear transformation, $e_i H M$, and we end up with a more expensive version of Hash Embeddings. See fig. 8b for the details.

**Tensor Train** (Yin et al., 2021) doesn't use hashing, but like QR it splits the input in a deterministic way that can be generalized to a random hash function if so inclined. Instead of adding or concatenating chunks, Tensor Train multiplies them together as matrices, which makes it not strictly a linear operation. However, like DHE, the first step in reducing the input size is some kind of sketching.

See Appendix C for comparisons to a few more related methods.

Our algorithm is the first that deviates from random sketching as the main vehicle of compression. However, as we replace it by *learned* sketching, we can still express the embedding process as $e_i H M$ for a sparse matrix $H$ and a small dense matrix $M$. The difference is that *we learn $H$ from the data* instead of using a random or fixed matrix. Experimentally and theoretically this allows better learning at the memory usage.

In section 4 we give the practical description of the algorithm, and in section 3 we analyze it mathematically in a simplified setting.

## 3 ANALYSIS

We consider a simplified model of the setup, where $X \in \mathbb{R}^{n \times d_1}$ and $Y \in \mathbb{R}^{n \times d_2}$, and we want to find the $T \in \mathbb{R}^{d_1 \times d_2}$ that minimizes $\|XT - Y\|_F^2$. In our setup $n > d_1 >> d_2$. (Here $\| \cdot \|_F$ is the Frobenius norm, defined by $\|X\|_F^2 = \sum_{i,j} X_{i,j}^2$.)

Obviously we can easily retrieve $T$ from $(X, Y)$ solving the least squares problem. Let's call this optimal solution $T^* \in \mathbb{R}^{d_1 \times d_2}$. However, we want to save memory, so we instead pick a sparse random matrix $H_0 \in \mathbb{R}^{d_1 \times 2k}$ and solve

$$M_0 = \underset{M \in \mathbb{R}^{2k \times d_2}}{\arg\min} \|XH_0M - Y\|_F^2.$$

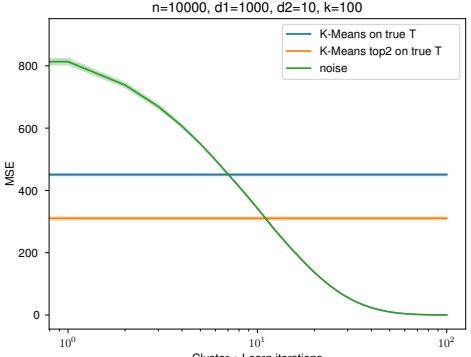
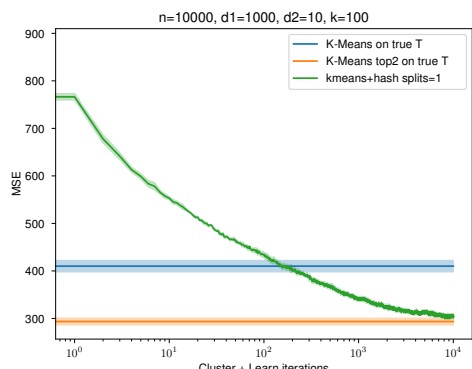

(a) **Dense CQR**: Repeatedly concatenating $T_i$ with random Gaussian noise quickly recovers the optimal $T^*$ solution to the least squares problem. This matches closely the exponential convergence we show in Theorem 1. For this plot we sampled $X \sim N(0,1)^{n \times d_1}$, $T \sim N(0,1)^{d_1 \times d_2}$ and let $Y = XT$. The result would be similar if $Y$ had noise added on top.

(b) **Multi-step CQR**: Repeatedly running $k$-means and concatenating with a Count Sketch finds a solution, $T_i$, that is as good as a $k$-means clustering of the optimal $T^*$ after just 100 iterations. If we continue training it eventually finds one that is as good as "top 2" $k$-means, even though it only uses normal $k$-means itself. We chose $X$ and $Y$ the same as on the left.

Figure 5: Empirically comparing the (slower) Dense CQR analyzed in this section with the (faster) K-Means approach in fig. 1.

Then $T_0 = H_0 M_0$ is an approximation to $T^*$, in so far as $H_0$ preserves the column space of $T^*$. Woodruff (2014) showed that this is indeed likely, if $k > d_2$ even if $H_0$ has just one non-zero per row. For technical reasons we will let $H_0$ be the horizontal concatenation of two Count Sketch matrices, so it has $2k$ columns with 2 non-zeros per row.

The idea is to improve over Count Sketch by analyzing $T_0$. The classical $k$-means algorithm can be seen to find the optimal factorization $T_0 \approx H'M'$, where $H' \in \{0,1\}^{d_1 \times k}$ has just one 1 per row. So we have saved a bit of space. We could stop here and declare victory, but we want to improve further. To do this we take $H_1 = [H' \mid C']$ – the concatenation of $H'$ with a new Count Sketch matrix. We can now repeat the optimization $M_1 = \arg\min_{M \in \mathbb{R}^{2k \times d_2}} \|XH_1 M - Y\|_F^2$ and rinse and repeat:

**Multi-step CQR:**   Let $H_0 = 0 \in \mathbb{R}^{d_1 \times 2k}$, $M_0 = 0 \in \mathbb{R}^{2k \times d_2}$; then repeat for $i \geq 0$:

$$T_i = H_i M_i$$
$$H_{i+1} = [k\text{-means}(T_i) \mid \text{new-count-sketch}()] \in \mathbb{R}^{d_1 \times 2k}.$$
$$M_{i+1} = \arg\min_M \|XH_{i+1}M - Y\|_F^2 \in \mathbb{R}^{2k \times d_2}.$$

Note that we only ever use the "assignment matrix" from $k$-means, not the actual cluster centers found by Lloyd's algorithm. One could use a different approximation to $T_i$'s column space instead, trading of the sparsity for convergence in fewer steps. Indeed we will now modify the algorithm to a dense version that is more amendable to theoretical analysis.

Instead of $k$-means and Count Sketch, define $H_{i+1} = [T_i \mid N(0,1)^{d_1 \times k}]$. That is, $H_i$ is the previous $T_i$ concatenated with a random normal matrix. In appendix B we show experimentally that this algorithm (which uses a dense $H_i$ rather than a sparse, and thus is slower than what we propose above) behaves nearly identically to the multi-step CQR.

We show that for this version of the algorithm:

**Theorem 1.**

$$E[\|XT_i - Y\|_F^2] \leq (1 - \rho)^{ik} \|XT^*\|_F^2 + \|XT^* - Y\|_F^2,$$

where $\rho = \|X\|_{-2}^2 / \|X\|_F^2 \approx 1/d_1$ is the smallest singular value of $X$ squared divided by the sum of singular values squared.[2] This means that after $i = O(\frac{d_1}{k} \log(\|XT^*\|_F/\varepsilon))$ iterations we have an $\varepsilon$ approximation to the optimal solution.

Note that the standard least squares problem can be solved in $O(nd_1d_2)$ time, but one iteration of our algorithm only takes $O(nkd_2)$ time. Repeating it for $d_1/k$ iterations is thus no slower than the default algorithm for the general least squares problem, but using less memory.

## 4 IMPLEMENTATION OF CLUSTERED COMPOSITIONAL EMBEDDINGS (CQR)

We now describe the CQR method as used in our experiments. The description here differs from fig. 1 in two main aspects: 1) We build 4 separate tables, and concatenate the output. And 2) We only do one iteration of the "learning, clustering, learning" loop.

For illustration purposes, we use the setup of one of our experiments on the Kaggle dataset, where the embedding dimension is 16 and the number of parameters is $16 \cdot 1000$.

**Step 1:** We start as in QR hybrid (see fig. 4b). with 2 times 4 tables $T_1, \ldots, T_4$ and $T'_1, \ldots, T'_4$, each of size $500 \times 4$. We also pick random hash functions $h_1, \ldots, h_4, h'_1, \ldots, h'_4 \colon [n] \to [500]$. Then the embedding of ID $i \in [n]$ is defined to be the concatenation

$$\begin{pmatrix} T_1[h_1(i)] & T_2[h_2(i)] & T_3[h_3(i)] & T_4[h_4(i)] \\ + & + & + & + \\ T'_1[h'_1(i)] & T'_2[h'_2(i)] & T'_3[h'_3(i)] & T'_4[h'_4(i)] \end{pmatrix}.$$

We then train the embedding tables for either half an epoch or a full epoch.

**Step 2:** Following fig. 1 we now cluster the large implicit table for each column $j = 1 \ldots 4$. We pick a sub-sample $S \subseteq [n]$ of $|S| = 256 \cdot k$ IDs, and compute the vectors for each of them to obtain a table $T_j^c$ of 1000 centroids. We define $h_j^c \colon [n] \to [1000]$ to send $i \in [n]$ to the nearest centroid of $T_j[h_j(i)] + T'_j[h'_j(i)]$.

**Step 3:** We retrain the model with the new learned hash function. The embedding vector of each $i \in [n]$ is now assigned to

$$\left( T_1^c[h_1^c(i)] \quad , \quad T_2^c[h_2^c(i)] \quad , \quad T_3^c[h_3^c(i)] \quad , \quad T_4^c[h_4^c(i)] \right).$$

Note this is similar to the QR concat method (fig. 4a) but with a learned hash function.

One may wonder about how best to store the learned hash functions $h_j^c$. We give more details on this in appendix E.

## 5 EXPERIMENTS

The main experimental finding is shown in fig. 6: The CQR method allows training a model with the baseline Binary Cross Entropy (BCE) using a third of the parameters needed with the hashing trick, and 1.5 times less than with the QR concat method. If given the optimal amount of parameters and trained till convergence, it can give a measurably lower BCE.

In this section we explain the experimental setup we used, and show the results of various other experiments highlighting the strengths and weaknesses of the CQR method.

### 5.1 EXPERIMENTAL SETUP

We follow the setup of the open-sourced DLRM code of Naumov et al. (2019).

**Dataset.** We use Criteo's Kaggle dataset and Criteo's Terabyte dataset of click logs. Both are public datasets and were used for benchmarking in DLRM (Naumov et al. (2019)), our backbone recommendation model. Each dataset has 13 dense features and 26 categorical features.

---

[2]In the appendix we also show how modify the algorithm to get $\rho = 1/d_1$ exactly.

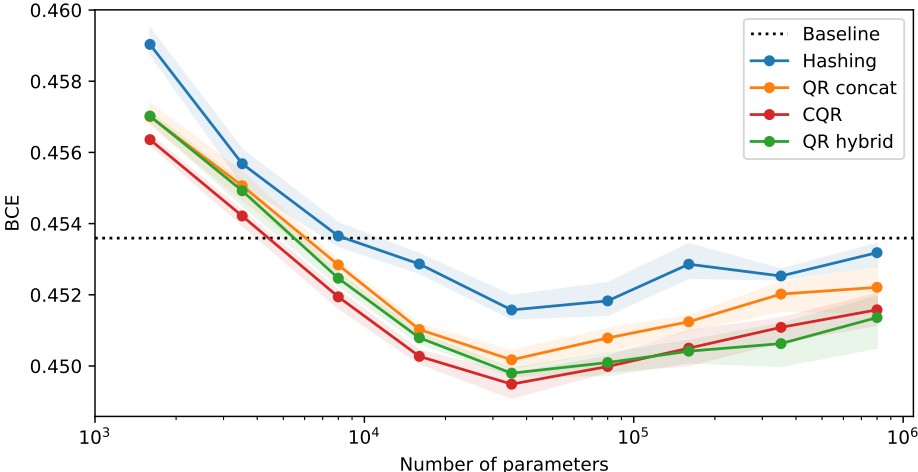

Figure 6: **Clustered QR outperforms Hashing and Compositional methods**. We trained DLRM on the Kaggle dataset with different table compression algorithms and different table sizes for 10 epochs and picked the best validation loss. Our method was able to reach the baseline (trained for 1 epoch with $16 \cdot 10^7$ parameters, see section 5.2) with just over 4000 total parameters. Meanwhile, the best previously published method (QR concat, Shi et al. (2020)) needed more than 6000. Our QR hybrid method also slightly outperform QR concat.

- The Kaggle dataset has about 45 million samples over 7 days, where each categorical feature has at most 10 million possible values.
- The Terabyte dataset has about 4 billion samples over 24 days. In our experiments, we sub-sampled one-eighth of the data and pre-hashed them with a simple modulus such that each categorical feature has at most 10 million possible values, as recommended by the default benchmarking setting in the DLRM code.

We use data from the days prior to the last day as the training set. The data of the last data is split into a validation set and a test set.

**Evaluation Metric.** We use Binary Cross-Entropy (BCE) to evaluate the performance of the recommendation model. This metric is also used in Shi et al. (2020) for evaluating the performance of the QR methods. A small BCE means the prediction probability is close to the actual result.

**Epochs.** Recommendation systems are typically just trained for one epoch since they are prone to over-fitting. It has however been observed (e.g. by Desai et al. (2022)) that compressed embedding tables benefit from multi-epoch training. We thus give results both for 1 epoch training, 2 epoch training and "best of 10 epochs" (in fig. 6), making sure we get the actual best possible model performance.

**Backbone Recommendation Models.** We adopt the deep learning recommendation model (DLRM) from Naumov et al. (2019) as the backbone recommendation model; see fig. 2. The DLRM takes dense and sparse features as inputs, where a full embedding table (i.e., one with $|V|$ many rows) is used for each categorical feature. Since the model has an open-sourced implementation, we only have to modify the embedding table part of the code to implement our methods.

**Number of runs of each model.** On the Kaggle dataset, PQ inference is run once, while all other models are run 3 times. On the Terabyte dataset, each model is run once.

## 5.2 BASELINE: FULL EMBEDDING TABLE

Full embedding table means that for each categorical feature, we train an embedding table of the size of the number of categorical values. Then a categorical value is mapped uniquely to a row in the table. For example, if a feature has 10 million possible values, then its embedding table would have

10 million row of embedding vectors. The embedding table is then trained for one epoch. However, using a full embedding table is usually not practical.

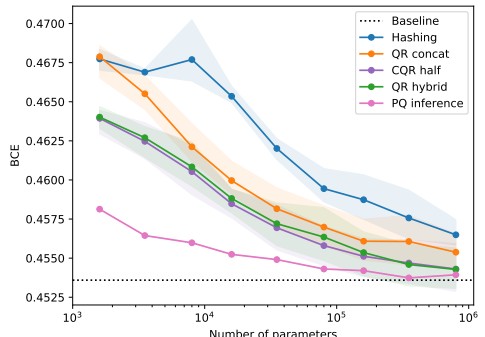
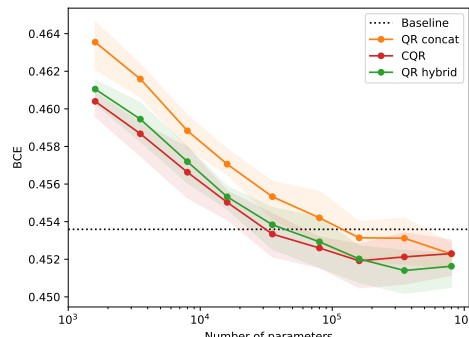

(a) **Kaggle dataset, 1 epoch**: Since CQR, as defined, clusters the table after the first epoch, we here suggest "CQR half" which clusters after only half an epoch. While this outperformed the previous methods (Hashing and QR concat) it was slightly worse than just keeping our own sketching method (QR hybrid) for the entire epoch.

(b) **Kaggle dataset, 2 epochs**: Training for two epochs is enough for CQR to outperform QR hybrid as well as the methods from previous papers, though not by as much as in fig. 6 where we trained till convergence.

Figure 7: All methods trained for 1 or 2 epochs on Kaggle

## 5.3 PQ INFERENCE

As described in the introduction, the CQR method mimics Product Quantization (PQ) on the full embedding table. For that reason, PQ is a strong baseline which we compare with in fig. 7a and fig. 8a. As it turns out we are mostly able to outperform this baseline, which while surprising may be explained by our PQ not being "finetuned" after clustering. We tried training a full table, PQ clustering it and training this clustering for an epoch, but this approach immediately overfit and gave terrible results. Interestingly this suggests CQR works as a regularization method for PQ.

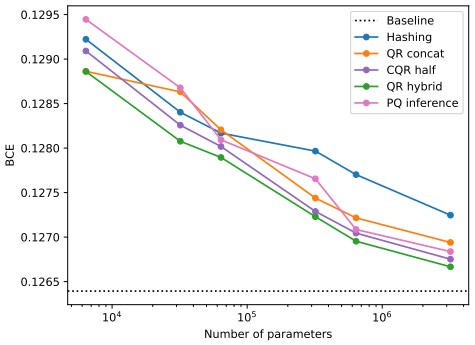
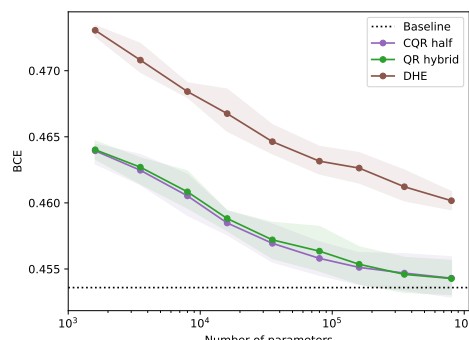

(a) **Terabyte dataset, 1 epoch**: Like on the Kaggle dataset, running for just 1 epoch was not enough to show an improvement over our sketching method, QR hybrid, but enough to beat QR concat and the Hashing method.

(b) **DHE, Kaggle dataset, 1 epoch**: Using the suggested MLP setup from Kang et al. (2021) we were not able to get competitive performance when fixing the number of layers to match the desired number of parameters. This is surprising, so we have opted to not include DHE on the other plots.

Figure 8: Experiments on Terabyte dataset and with Deep Hashing Embeddings.

# 6 CONCLUSION

We have shown the feasibility of compressing embedding tables at training time using clustering. While there is still work to be done in expanding our theoretical understanding and testing the method in more situations, we believe this shows an exciting new paradigm for dynamic sparsity in neural networks and recommender systems in particular.

Some previous papers have made wildly different claims in terms of the possibility of compressing embedding tables. We believe fig. 7a, fig. 7b and fig. 6 help explain some of that confusion: With standard learning rates, it is not possible to improve over the DLRM baseline much, if at all, at 1 epoch of training. This matches the results of Naumov et al. (2019). However, if you train till convergence, you can match the baseline with 1000x times fewer parameters than the default. Even if you just use the simple hashing trick. This matches the results of Desai et al. (2022).

In all experiments, except fig. 8a on the Terabyte dataset, our method, CQR, outperforms all other methods we tried. We believe it likely would be better on the Terabyte dataset as well, if given time to train until convergence.

Training to convergence is however not very common in practical recommendation systems. Our experiment in fig. 6 may offer an explanation why: Most methods overfit if given too large embedding tables(!) This is quite shocking, since the common knowledge is that bigger is always better and concepts should ideally have their own private embedding vectors. We believe these experiments call for more investigation into overfitting in DLRM style models.

## REPRODUCIBILITY STATEMENT

The backbone recommendation model, DLRM by Naumov et al. (2019), has an open-sourced PyTorch implementation of the model available on github, with instructions on how to use them in the benchmarking folder. It also has instructions on how to download the public datasets. We follow very closely to their instructions, so reproducing the baseline result should be straightforward.

For the QR methods, we only need to change two functions in the code: `create_emb` and `apply_emb`. We suggest using a class for each QR method; see fig. 4. For the random hash function, one could use a universal hash function or `numpy.random.randint`.

For the CQR method, apart from using the classes for the QR concat method and the QR hybrid method, we also need to add a section in the code for $k$-means clustering after some training. We recommend using the open-sourced $k$-means clustering algorithm by Johnson et al. (2019), since it is much faster than Scikit-learn's $k$-means clustering algorithm (Pedregosa et al., 2011).

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

CONTENTS

## A  WHAT DIDN'T WORK

Here are the ideas we tried but didn't work at the end.

**Using multiple helper tables**  It is a natural idea use more than one helper table. However, in our experiments, the effect of having more helper tables is not apparent.

**Circular clustering**  Based on the QR concat method, the circular clustering method would use information from other columns to do clustering. However, the resulting index pointer functions are too similar to each other, meaning that this method is essentially the hashing trick. We further discuss this issue in appendix G.

**Continuous clustering**  We originally envisioned our methods in a tight loop between training and (re)clustering. It turned out that reducing the number of clusterings didn't impact performance, so we eventually reduced it all the way down to just one. In practical applications, with distribution shift over time, doing more clusterings may still be useful, as we discuss in section 3.

**Changing the number of columns** In general, increasing the number of columns leads to better results. However the marginal benefits quickly decrease, and as the number of hash functions grow, so does the training and inference time. We found that 4 columns / hash-functions was a good spot.

**Residual vector quantization** The CQR method combines Product Quantization (PQ) with the QR concat method. We tried combining Residual vector quantization (RVQ) with the Hash Embeddings method from Tito Svenstrup et al. (2017). This method does not perform significantly better than the Hash Embeddings method.

**Seeding with PQ** We first train a full embedding table for one epoch, and then do Product Quantization (PQ) on the table to obtain the index pointer functions.

We then use the index pointer functions instead of random hash functions in the QR concat method. This method turned out performing badly: The training loss quickly diverges from the test loss after training on just a few batches of data.

Here are some variations of the CQR method:

**Earlier clustering** We currently have two versions of the CQR method: CQR half, where clustering happens at the middle of the first epoch, and CQR, where clustering happens at the end of the first epoch. We observe that when we cluster earlier, the result is slightly worse. Though in our case the CQR half method still outperforms the QR concat method.

**More parameters before clustering** The CQR method allows using two number of parameters, one in Step 1 where we follow the QR hybrid method to get a sketch, and one in Step 3 where we follow the QR concat method. We thought that by using more parameters at the beginning, we would be able to get a better set of index pointer tables. However, the experiment suggested that the training is faster but the terminal performance is not significantly better.

## B  PROOF OF THE MAIN THEOREM

Let's remind ourselves of the "Dense CQR algorithm" from section 3: Given $X \in \mathbb{R}^{n \times d_1}$ and $Y \in \mathbb{R}^{n \times d_2}$, pick $k$ such that $n > d_1 > k > d_2$. We want to solve find a matrix $T^*$ of size $d_1 \times d_2$ such that $\|XT^* - Y\|_F$ is minimized – the classical Least Squares problem. However, we want to use memory less than the typical $nd_1^2$. We thus use this algorithm:

**Dense CQR Algorithm:** Let $T_0 = 0 \in \mathbb{R}^{d_1 \times d_2}$. For $i = 1$ to $m$:

$$\text{Sample} \quad G_i \sim N(0,1)^{d_1 \times (k-d_2)};$$
$$\text{Compute} \quad H_i = [T_{i-1} \mid G_i] \in \mathbb{R}^{d_1 \times k}$$
$$M_i = \arg\inf_M \|XH_iM - Y\|_F^2 \in \mathbb{R}^{k \times d_2}.$$
$$T_i = H_iM_i$$

We will now argue that $T_m$ is a good approximation to $T^*$ in the sense that $\|XT_m - Y\|_F^2$ is not much bigger than $\|XT^* - Y\|_F^2$.

Let's consider a non-optimal choice of $M_i$ first. Suppose we set $M_i = \begin{bmatrix} I_{d_2} \\ M_i' \end{bmatrix}$ where $M_i'$ is chosen such that $\|H_iM_i - T^*\|_F$ is minimized. By direct multiplication, we have $H_iM_i = T_{i-1} + G_iM_i'$. Hence in this case minimizing $\|H_iM_i - T^*\|_F$ is equivalent to finding $M_i'$ at each step such that $\|G_iM_i' - (T^* - T_{i-1})\|_F$ is minimized.

In other words, we are trying to estimate $T^*$ with $\sum_i G_iM_i'$, where each $G_i$ is random and each $M_i'$ is greedily chosen at each step. This is similar to, for example, the approaches in Barron et al. (2008), though they use a concrete list of $G_i$'s. In their case, by the time we have $d_1/k$ such $G_i$'s, we are just multiplying $X$ with a $d_1 \times d_1$ random Gaussian matrix, which of course will have full rank, and so the concatenated $M$ matrix can basically ignore it. However, in our case we do a local, not global optimization over the $M_i$.

Recall the theorem:

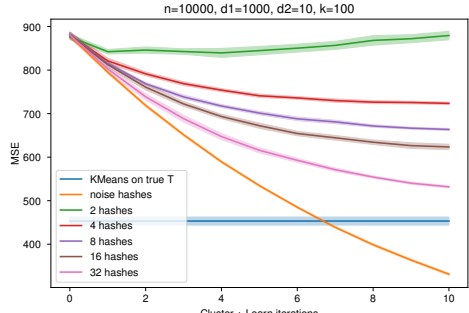 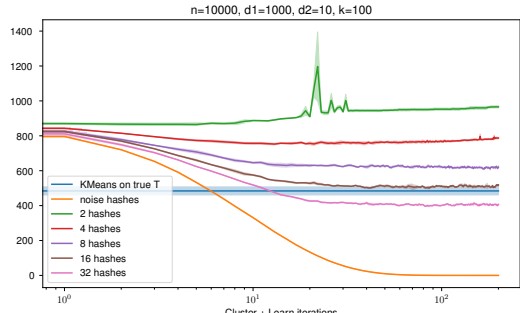

(a) **Multi-step CQR with 10 steps**: All methods, dense as sparse, get nearly the same error after the zero steps. This means we haven't done any clustering, but simply run a more or less sparse Count Sketch. At 1 step all sparsities still do quite well, which is why the real CQR method proposed in this paper uses just 2 hash functions per column.

(b) **Multi-step CQR with 200 steps**: All sparse methods converge before reaching zero loss. This is expected, since the $k$-means baseline run on the true, optimal $T^*$ matrix also doesn't achieve zero loss. Meanwhile, the dense method analyzed in this section does eventually achieve zero loss as predicted by the theorem.

Interestingly $k$-means stopped being an effective CQR sparsification algorithm at higher number of steps, and we here instead assign each ID to the nearest codewords from a random Gaussian codebook.

Figure 9: **Multi-step CQR on Least Squares**: We compare the Multi-step CQR method with sparse $H$ ($n$ hashes per row) with the dense model analyzed in theorem 1. We also show a baseline of first solving for the optimal $T^*$ and then running $k$-means to reduce the space usage. As hoped, and expected, they all quickly improve with more iterations, though the more sparse version eventually converge.

**Theorem 1.** *Given $X \in \mathbb{R}^{n \times d_1}$ and $Y \in \mathbb{R}^{n \times d_2}$. Let $T^* = \arg\min_{T \in \mathbb{R}^{d_1 \times d_2}} \|XT - Y\|_F^2$ be an optimal solution to the least squares problem. Then*

$$\mathrm{E}\left[\|XT_i - Y\|_F^2\right] \le (1-\rho)^{i(k-d_2)}\|XT^*\|_F^2 + \|XT^* - Y\|_F^2,$$

*where $\rho = \|X\|_{-2}^2/\|X\|_F^2$.*

Here we use the notation that $\|X\|_{-2}$ is the smallest singular value of $X$.

**Corollary 1.** *In the setting of the theorem, if all singular values of $X$ are equal, then*

$$\mathrm{E}\left[\|XT_i - Y\|_F^2\right] \le e^{-i\frac{k-d_2}{d_1}}\|XT^*\|_F^2 + \|XT^* - Y\|_F^2.$$

*Proof of Corollary 1.* Note that $\|X\|_F^2$ is the sum of the $d_1$ singular values squared: $\|X\|_F^2 = \sum_i \sigma_i^2$. Since all singular values are equal, say to $\sigma \in \mathbb{R}$, then $\|X\|_F^2 = d_1\sigma^2$. Similarly in this setting, $\|X\|_{-2}^2 = \sigma^2$ so $\rho = 1/d_1$. Using the inequality $1 - 1/d_1 \le e^{-1/d_1}$ gives the corollary. $\qquad\square$

*Proof of Theorem 1.* First split $Y$ into the part that's in the column space of $X$ and the part that's not, $Z$. We have $Y = XT^* + Z$, where $T^* = \arg\min_T \|XT - Y\|_F$ is the solution to the least squares problem. By Pythagoras theorem we then have

$$\mathrm{E}\left[\|XT_i - Y\|_F^2\right] = \mathrm{E}\left[\|XT_i - (XT^* + Z)\|_F^2\right] = \mathrm{E}\left[\|X(T_i - T^*)\|_F^2\right] + \|Z\|_F^2,$$

so it suffices to show

$$\mathrm{E}\left[\|X(T_i - T^*)\|_F^2\right] \le (1-\rho)^{i(k-d_2)}\|XT^*\|_F^2.$$

We will prove the theorem by induction over $i$. In the case $i = 0$ we have $T_i = 0$, so $\mathrm{E}[\|X(T_0 - T^*\|_F^2] = \mathrm{E}[\|XT^*\|_F^2]$ trivially. For $i \geq 1$ we insert $T_i = H_i M_i$ and optimize over $M_i'$:

$$
\begin{aligned}
\mathrm{E}[\|X(T_i - T^*)\|_F^2] &= \mathrm{E}[\|X(H_i M_i - T^*)\|_F^2] \\
&\leq \mathrm{E}[\|X(H_i[I \mid M_i'] - T^*)\|_F^2] \\
&= \mathrm{E}[\|X((T_{i-1} + G_i M_i') - T^*)\|_F^2] \\
&= \mathrm{E}[\|X(G_i M_i' - (T^* - T_{i-1}))\|_F^2]. \\
&= \mathrm{E}[\mathrm{E}[\|X(G_i M_i' - (T^* - T_{i-1}))\|_F^2 \mid T_{i-1}]] \\
&\leq (1 - \rho)^{k-d_2} \mathrm{E}[\|X(T^* - T_{i-1})\|_F^2] \\
&\leq (1 - \rho)^{i(k-d_2)} \|XT^*\|_F^2,
\end{aligned}
$$

where the last step followed by induction. The critical step here was bounding

$$
\mathrm{E}_G[\inf_M \|X(GM - T)\|_F^2] \leq (1 - \rho)^{k-d_2} \|XT\|_F^2,
$$

for a fixed $T$. We will do this in a series of lemmas below. $\qquad\square$

We show the lemma first in the "vector case", corresponding to $k = 2, d_2 = 1$. The general matrix case follow below, and is mostly a case of induction on the vector case.

**Lemma 1.** *Let $X \in \mathbb{R}^{n \times d}$ be a matrix with singular values $\sigma_1 \geq \cdots \geq \sigma_d \geq 0$. Define $\rho = \sigma_d^2 / \sum_i \sigma_i^2$, then for any $t \in \mathbb{R}^d$,*

$$
\mathrm{E}_{g \sim N(0,1)^d} \left[ \inf_{m \in \mathbb{R}} \|X(gm - t)\|_2^2 \right] \leq (1 - \rho) \|Xt\|_2^2.
$$

*Proof.* Setting $m = \langle Xt, Xg \rangle / \|Xg\|_2^2$ we get

$$
\|X(gm - t)\|_2^2 = m^2 \|Xg\|_2^2 + \|Xt\|_2^2 - 2m\langle Xg, Xt \rangle \tag{1}
$$

$$
= \left(1 - \frac{\langle Xt, Xg \rangle^2}{\|Xt\|_2^2 \|Xg\|_2^2}\right) \|Xt\|_2^2. \tag{2}
$$

We use the singular value decomposition of $X = U\Sigma V^T$. Since $g \sim N(0,1)^d$ and $V^T$ is unitary, we have $V^T g \sim N(0,1)^d$ and hence we can assume $V = I$. Then

$$
\frac{\langle Xt, Xg \rangle^2}{\|Xt\|_2^2 \|Xg\|_2^2} = \frac{(t^T \Sigma U^T U \Sigma g)^2}{\|U\Sigma t\|_2^2 \|U\Sigma g\|_2^2} \tag{3}
$$

$$
= \frac{(t^T \Sigma^2 g)^2}{\|\Sigma t\|_2^2 \|\Sigma g\|_2^2} \tag{4}
$$

$$
= \frac{(\sum_i t_i \sigma_i^2 g_i)^2}{(\sum_i \sigma_i^2 t_i^2)(\sum_i \sigma_i^2 g_i^2)}, \tag{5}
$$

where eq. (4) follows from $U^T U = I$ in the SVD. We expand the upper sum to get

$$
\mathrm{E}_g \left[ \frac{(\sum_i t_i \sigma_i^2 g_i)^2}{(\sum_i \sigma_i^2 t_i^2)(\sum_i \sigma_i^2 g_i^2)} \right] = \mathrm{E}_g \left[ \frac{\sum_{i,j} t_i t_j \sigma_i^2 \sigma_j^2 g_i g_j}{(\sum_i \sigma_i^2 t_i^2)(\sum_i \sigma_i^2 g_i^2)} \right] \tag{6}
$$

$$
= \mathrm{E}_g \left[ \frac{\sum_i t_i^2 \sigma_i^4 g_i^2}{(\sum_i \sigma_i^2 t_i^2)(\sum_i \sigma_i^2 g_i^2)} \right]. \tag{7}
$$

Here we use the fact that the $g_i$'s are symmetric, so the cross terms of the sum have mean 0. By scaling, we can assume $\sum_i \sigma_i^2 t_i^2 = 1$ and define $p_i = \sigma_i^2 t_i^2$. Then the sum is just a convex combination:

$$
(7) = \sum_i p_i \mathrm{E}_g \left[ \frac{\sigma_i^2 g_i^2}{\sum_i \sigma_i^2 g_i^2} \right]. \tag{8}
$$

Since $\sigma_i \geq \sigma_d$ and $g_i$'s are IID, by direct comparison we have

$$\mathrm{E}_g\left[\frac{\sigma_i^2 g_i^2}{\sum_i \sigma_i^2 g_i^2}\right] \geq \mathrm{E}_g\left[\frac{\sigma_d^2 g_d^2}{\sum_i \sigma_i^2 g_i^2}\right]$$

Hence

$$(7) \geq \mathrm{E}_g\left[\frac{\sigma_d^2 g_d^2}{\sum_i \sigma_i^2 g_i^2}\right]\sum_i p_i = \mathrm{E}_g\left[\frac{\sigma_d^2 g_d^2}{\sum_i \sigma_i^2 g_i^2}\right].$$

It remains to bound

$$\mathrm{E}_g\left[\frac{\sigma_d^2 g_d^2}{\sum_i \sigma_i^2 g_i^2}\right] \geq \frac{\sigma_d^2}{\sum_i \sigma_i^2} = \rho, \tag{9}$$

but this follows from a cute, but rather technical lemma, which we will postpone to the end of this section. (lemma 3.) $\square$

It is interesting to notice how the improvement we make each step (that is $1 - \rho$) could be increased to $1 - 1/d$ by picking $G$ from a distribution other than IID normal.

If $X = U\Sigma V^T$, we can also take $g = V\Sigma^{-1}g'$, where $g' \sim N(0,1)^{d_1 \times (k - d_2)}$. In that case we get

$$\mathrm{E}\left(\frac{\langle Xt, Xg\rangle}{\|Xg\|_2\|Xt\|_2^2}\right)^2 = \mathrm{E}\left(\frac{t^T V\Sigma^2 V^T g}{\|Ug'\|_2\|Xt\|_2^2}\right)^2 = \mathrm{E}\left(\frac{t^T V\Sigma g'}{\|g'\|_2\|Xt\|_2^2}\right)^2 = \frac{1}{d_1}\frac{\|t^T V\Sigma\|_2^2}{\|Xt\|_2^2} = \frac{1}{d_1}.$$

So this way we recreate the ideal bound from corollary 1. Note that $\frac{\|X\|_{-2}^2}{\|X\|_F^2} \leq 1/d_1$. Of course it comes with the negative side of having to compute the SVD of $X$. But since this is just a theoretical algorithm, it's still interesting and shows how we would ideally update $T_i$. See fig. 10 for the effect of this change experimentally.

It's an interesting problem how it might inspire a better CQR algorithm. Somehow we'd have to get information about the the SVD of $X$ into our sparse super-space approximations.

We now show how to extend the vector case to general matrices.

**Lemma 2.** *Let $X \in \mathbb{R}^{n \times d_1}$ be a matrix with singular values $\sigma_1 \geq \cdots \geq \sigma_{d_1} \geq 0$. Define $\rho = \sigma_{d_1}^2 / \sum_i \sigma_i^2$, then for any $T \in \mathbb{R}^{n \times d_2}$,*

$$\mathrm{E}_{G \sim N(0,1)^{d_1 \times k}}\left[\inf_{M \in \mathbb{R}^{k \times d_2}} \|X(GM - T)\|_F^2\right] \leq (1 - \rho)^k \|XT\|_F^2.$$

*Proof.* The case $k = 1, d_2 = 1$ is proven above in lemma 1.

**Case $k = 1$:** We first consider the case where $k = 1$, but $d_2$ can be any positive integer (at most $k$). Let $T = [t_1|t_2|\dots|t_{d_2}]$ be the columns of $T$ and $M = [m_1|m_2|\dots|m_{d_2}]$ be the columns of $M$. Then the $i$th column of $X(GM - T)$ is $X(Gm_i - t_i)$, and since the squared Frobenius norm of a matrix is simply the sum of the squared column l2 norms, we have

$$\mathrm{E}[\|X(GM - T)\|_F^2] = \mathrm{E}\left[\sum_{i=1}^{d_2} \|X(Gm_i - t_i)\|_2^2\right]$$

$$= \sum_{i=1}^{d_2} \mathrm{E}\left[\|X(Gm_i - t_i)\|_2^2\right]$$

$$\leq \sum_{i=1}^{d_2}(1 - \rho)\mathrm{E}[\|Xt_i\|_2^2] \tag{10}$$

$$= (1 - \rho)\mathrm{E}\left[\sum_{i=1}^{d_2} \|Xt_i\|_2^2\right]$$

$$= (1 - \rho)\mathrm{E}[\|XT\|_F^2].$$

where in (10) we applied the single vector case.

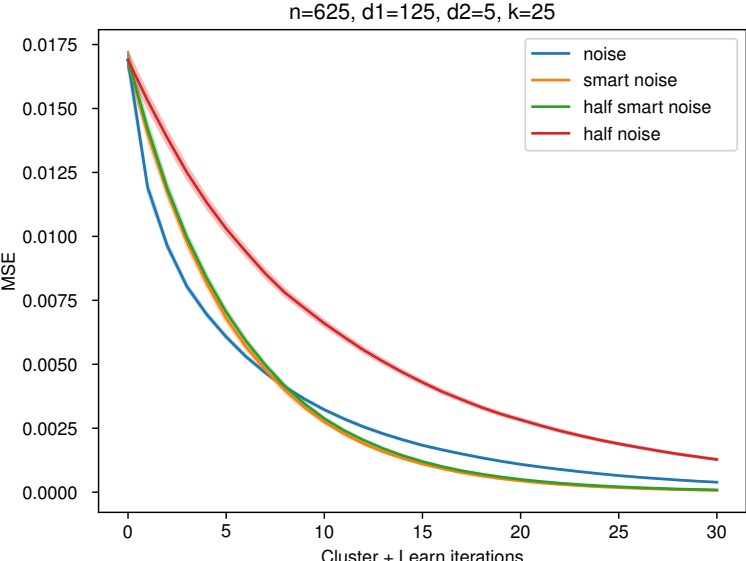

Figure 10: **SVD aligned noise converges faster.** In the discussion we mention that picking the random noise in $H_i$ as $g = V\Sigma^{-1}g'$, where $g' \sim N(0,1)^{d_1 \times (k-d_2)}$, can improve the convergence rate from $(1-\rho)^{ik}$ to $(1-1/d)^{ik}$, which is always better. In this graph we experimentally compare this approach (labeled "smart noise") against the IID gaussian noise (labeled "noise"), and find that the smart noise indeed converges faster – at least once we get close to zero noise. The graph is over 40 repetitions where $X$ is a random rank-10 matrix plus some low magnitude noise.

We also investigate how much we lose in the theorem by only considering $M$ on the form $[I|M']$, rather than a general $M$ that could take advantage of last rounds $T_i$. The plots labeled "half noise" and "half smart noise" are limited in this way, while the two others are not. We observe that the effect of this is much larger in the "non-smart" case, which indicates that the optimal noise distribution we found might accidentally be tailored to our analysis.

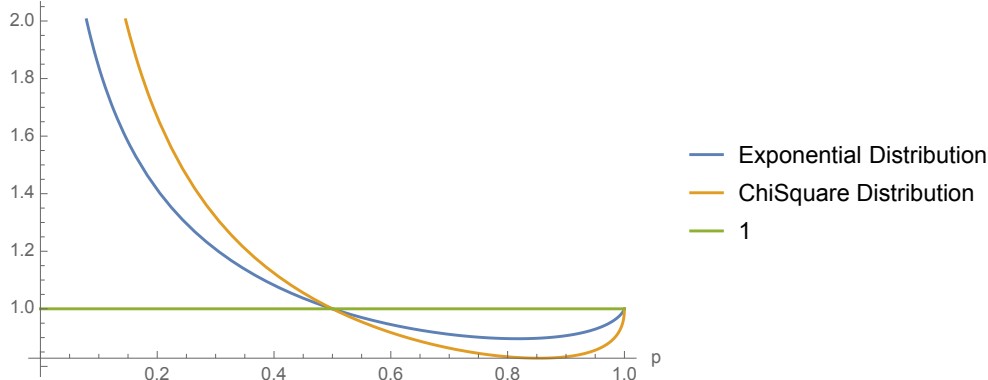

Figure 11: Expectation, $\mathrm{E}\left[\frac{x}{px+(1-p)y}\right]$ when $x, y$ are IID with Exponential (blue) or Chi Square distribution (Orange). In both cases the expectation is $\geq 1$ when $p \leq 1/2$, just as lemma 3 predicts.

**Case $k > 1$:** This time, let $g_1, g_2, \ldots, g_k$ be the columns of $G$ and let $m_1^T, m_2^T, \ldots, m_k^T$ be the *rows* of $M$.

We prove the lemma by induction over $k$. We already proved the base-case $k = 1$, so all we need is the induction step. We use the expansion of the matrix product $GM$ as a sum of outer products $GM = \sum_{i=1}^{k} g_i m_i^T$:

$$
\begin{aligned}
\mathrm{E}[\|X(GM - T)\|_F^2] &= \mathrm{E}\left[\left\|X\left(\sum_{i=1}^{k} g_i m_i^T - T\right)\right\|_F^2\right] \\
&= \mathrm{E}\left[\left\|X\left(g_1 m_1^T + \left(\sum_{i=2}^{k} g_i m_i^T - T\right)\right)\right\|_F^2\right] \\
&\leq (1 - \rho)\,\mathrm{E}\left[X\left(\left\|\sum_{i=2}^{k} g_i m_i^T - T\right\|_F^2\right)\right] \quad (11) \\
&\leq (1 - \rho)^k\,\mathrm{E}\left[\|XT\|_F^2\right].
\end{aligned}
$$

where (11) used the $k = 1$ case shown above, and (12) used the inductive hypothesis. This completes the proof for general $k$ and $d_2$ that we needed for the full theorem. $\qquad\square$

## B.1 TECHNICAL LEMMAS

It remains to show an interesting lemma used for proving the vector case in lemma 1.

**Lemma 3.** *Let $a_1 \ldots, a_n \geq 0$ be IID random variables and assume some values $p_i \geq 0$ st. $\sum_i p_i = 1$ and $p_n \leq 1/n$. Then*

$$
E\left[\frac{a_n}{\sum_i p_i a_i}\right] \geq 1.
$$

This completes the original proof with $p_i = \frac{\sigma_i^2}{\sum_j \sigma_j^2}$ and $a_i = g_i^2$.

*Proof.* Since the $a_i$ are IID, it doesn't matter if we permute them. In particular, if $\pi$ is a random permutation of $\{1, \ldots, n-1\}$,

$$E\left[\frac{a_n}{\sum_i p_i a_i}\right] = E_a\left[E_\pi\left[\frac{a_n}{p_n a_n + \sum_i p_i a_{\pi_i}}\right]\right] \tag{12}$$

$$\geq E_a\left[\frac{a_n}{E_\pi\left[p_n a_n + \sum_{i<n} p_i a_{\pi_i}\right]}\right] \tag{13}$$

$$= E_a\left[\frac{a_n}{p_n a_n + \sum_{i<n} p_i\left(\frac{1}{n-1}\sum_{j<n} a_j\right)}\right] \tag{14}$$

$$= E_a\left[\frac{a_n}{p_n a_n + (1-p_n)\sum_{i<n}\frac{a_i}{n-1}}\right], \tag{15}$$

where eq. (13) uses Jensen's inequality on the convex function $1/x$.

Now define $a = \sum_{i=1}^n a_i$. By permuting $a_n$ with the other variables, we get:

$$E_a\left[\frac{a_n}{p_n a_n + (1-p_n)\sum_{i<n}\frac{a_i}{n-1}}\right] = E_a\left[\frac{a_n}{p_n a_n + \frac{1-p_n}{n-1}(a-a_n)}\right] \tag{16}$$

$$= E_a\left[\frac{1}{n}\sum_{i=1}^n \frac{a_i}{p_n a_i + \frac{1-p_n}{n-1}(a-a_i)}\right] \tag{17}$$

$$= E_a\left[\frac{1}{n}\sum_{i=1}^n \frac{a_i/a}{\frac{1-p_n}{n-1} - \left(\frac{1-p_n}{n-1} - p_n\right)a_i/a}\right] \tag{18}$$

$$= E_a\left[\frac{1}{n}\sum_{i=1}^n \phi(a_i/a)\right], \tag{19}$$

where

$$\phi(q_i) = \frac{q_i}{\frac{1-p_n}{n-1} - \left(\frac{1-p_n}{n-1} - p_n\right)q_i}$$

is convex whenever $\frac{1-p_n}{n-1}\big/\left(\frac{1-p_n}{n-1} - p_n\right) = \frac{1-p}{1-np} > 1$, which is true when $0 \leq p_n < 1/n$. That means we can use Jensen's again:

$$\frac{1}{n}\sum_{i=1}^n \phi(a_i/a) \geq \phi\left(\frac{1}{n}\sum_i \frac{a_i}{a}\right) = \phi\left(\frac{1}{n}\right) = 1,$$

which is what we wanted to show. $\qquad\square$

## C  OTHER RELATED WORK

Recent parallel work by Ghaemmaghami et al. (2022) presents a different way to learn a clustering without having to learn the exact embedding tables first. Instead of using hashing as a "pseudo table" for clustering to work on, they train a model with a lower dimensional table. They run a special clustering algorithm designed for low dimensional data on this table, and then retrain the model with the clustering known. It is interesting future work to better compare the strengths and weaknesses of our two methods against each other.

Other people have looked at "Learning CountSketch" such as Liu et al. (2020) who gave a formulation in the learning-based sketching paradigm proposed by Indyk et al. (2019). However, these algorithms require repeatedly retraining the model, which for big recommendation systems would be much too slow.

## D  HASHING

If $h : [n] \to [m]$ and $s : [n] \to \{-1, 1\}$ are random functions, a Count Sketch is a matrix $H \in \{0, -1, 1\}^{m \times n}$ where $H_{i,j} = s(i)$ if $h(i) = j$ and 0 otherwise. Charikar et al. (2002) showed that if $m$ is large enough, the matrix $H$ is a dimensionality reduction in the sense that the norm $\|x\|_2$ of any vector in $\mathbb{R}^n$ is approximately preserved, $\|Hx\|_2 \approx \|x\|_2$.[3]

This gives a simple theoretical way to think about the algorithms above: The learned matrix $T' = H^T T$ is simply a lower dimensional approximation to the real table that we wanted to learn. While the theoretical result requires the random "sign function" $s$ for the approximation to be unbiased, in practice this appears to not be necessary when directly learning $T'$. Maybe because the vectors can simply be slightly shifted to debias the result.

There are many strong theoretical results on the properties of Count Sketches. For example, Woodruff (2014) showed that they are so called "subspace embeddings" which means the dimensionality reduction is "robust" and doesn't have blind spots that SGD may accidentally walk into. However, the most practical result is that one only needs $h$ to be a "universal hash function" ala Carter & Wegman (1977), which can be as simple and fast as the "multiply shift" hash function by Dietzfelbinger et al. (1997).

If Count Sketch shows that hashing each $i \in [n]$ to a single row in $[m]$, we may wonder why methods like Hash Embeddings use multiple hash functions (or DHE uses more than a thousand.) The answer can be seen in the theoretical analysis of the "Johnson Lindenstrauss" transformation and in particular the "Sparse Johnson Lindenstrauss" as analyzed by Cohen et al. (2018). The analysis shows that if the data being hashed is not very uniform, it is indeed better to use more than one hash function (more than 1 non-zero per column in $H$.) The exact amount depends on characteristics in the data distribution, but one can always get away with a sparsity of $\epsilon$ when looking for a $1 + \epsilon$ dimensionality reduction. Hence we speculate that DHE could in general replace the 1024 hash functions with something more like Hash Embeddings with an MLP on top. Another interesting part of the Cohen et al. (2018) analysis is that one should ideally split $[m]$ in segments, and have one hash function into each segment. This matches the implementations we based our work on below.

## E  HOW TO STORE THE HASH FUNCTIONS

We note that unlike the random hash functions used in Step 1, the index pointer functions obtained from clustering takes space *linear in the amount of training data* or at least in the ID universe size. At first this may seem like a major downside of our method, and while it isn't different from the index tables needed after Product Quantization, it definitely is something extra not needed by purely sketching based methods.

We give three reasons why storing this table is not an issue in practice:

1. The index pointer functions can be stored on the CPU rather than the GPU, since they are used as the first step of the model before the training/inference data has been moved from the CPU to the GPU. Furthermore the index lookup is typically done faster on CPUs, since it doesn't involve any dense matrix operations.

2. The index pointers can replace the original IDs. Unless we are working in a purely streaming setting, the training data has to be stored somewhere. If IDs are 64 bit integers, replacing them with four 16-bit index pointers is net neutral.

3. Some hashing and pruning can be used as a prepossessing step, reducing the universe size of the IDs and thus the size of the index table needed.

## F  GRAPHS USING AUC AS THE METRIC

We also evaluate the models using AUC, another often employed metric for gauging the effectiveness of a recommendation model. For example, it was used in (Kang et al., 2021). It provides the

---

[3]This also implies that inner products are approximately preserved by the dimensionality reduction.

probability of getting a correct prediction when evaluating a test sample from a balanced dataset. Thus, a better model is implied by a larger AUC. In this section, we plot the graphs again using AUC.

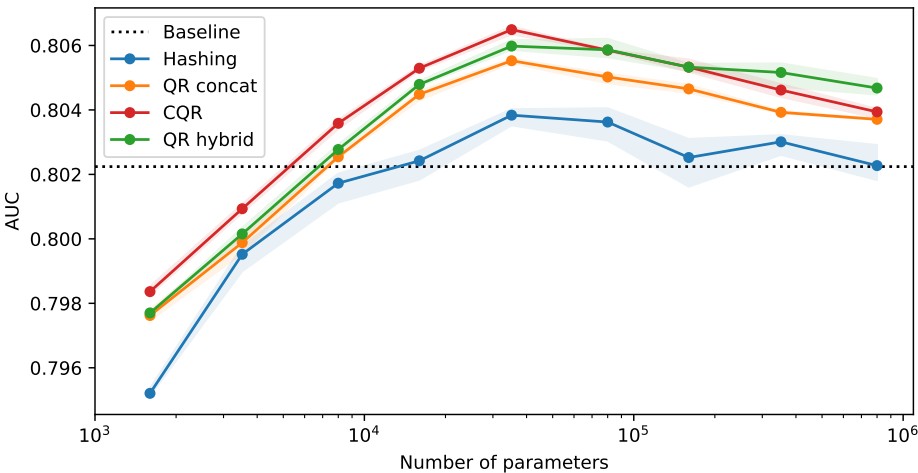

Figure 12: **Clustered QR outperforms Hashing and Compositional methods**. This figure is fig. 6 plotted with AUC.

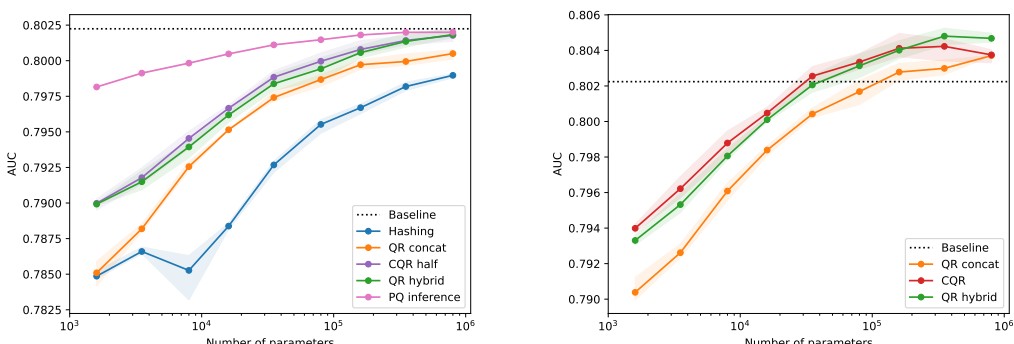

(a) **Kaggle dataset, 1 epoch**: This figure is fig. 7a plotted with AUC.

(b) **Kaggle dataset, 2 epochs**: This is fig. 7b plotted with AUC.

Figure 13: All methods trained for 1 or 2 epochs on Kaggle using AUC as the metric.

## G  TABLE COLLAPSE

Table collapsing was a problem we encountered for the circular clustering method as described in appendix A. We describe the problem and the metric we used to detect it here, since we think they may be of interest to the community.

Suppose we are doing $k$-means clustering on a table of 3 partitions in order to obtain 3 index pointer functions $h_j^c$. These functions can be thought as a table, where the $(i, j)$-entry is given by $h_j^c(i)$.

There are multiple failure modes we have to be aware of. The first one is column-wise collapse:

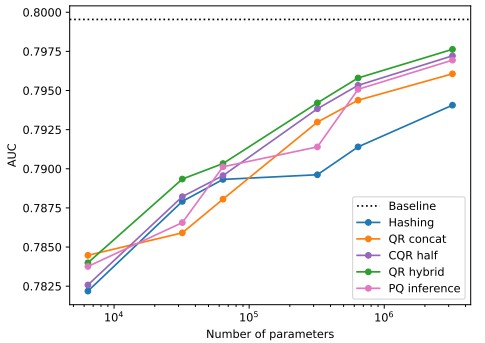 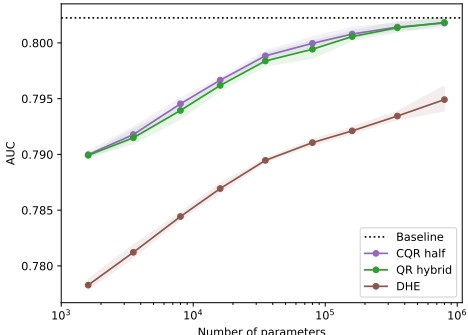

(a) **Terabyte dataset, 1 epoch**: This figure is fig. 8a plotted with AUC.

(b) **DHE, Kaggle dataset, 1 epoch**: This figure is fig. 8b plotted with AUC.

Figure 14: Experiments on Terabyte dataset and with Deep Hashing Embeddings using AUC as the metric.

| 1 | 0 | 0 |
|---|---|---|
| 1 | 1 | 2 |
| 1 | 0 | 3 |
| ⋮ | ⋮ | ⋮ |
| 1 | 3 | 1 |

In this table the first column has collapsed to just one cluster. Because of the way $k$-means clustering works, this exact case of complete collapse isn't actually possible, but we might get arbitrarily low entropy as measured by $H_1$, which we define as follows: For each column $j$, its column entropy is defined to be the entropy of the probability distribution $p_j \colon h_j^c([n]) \to [0, 1]$ defined by

$$p_j(x) = \frac{\#\{i : h_j^c(i) = x\}}{n}.$$

Then we define $H_1$ to be the minimum entropy of the (here 3) column-entropies.

The second failure mode is pairwise collapse:

| 1 | 0 | 1 |
|---|---|---|
| 2 | 2 | 3 |
| 1 | 0 | 3 |
| 3 | 1 | 0 |
| 2 | 2 | 1 |

In this case the second column is just a permutation of the first column. This means the expanded set of possible vectors is much smaller than we would expect. We can measure pairwise collapse by computing the entropy of the histogram of pairs, where the entropy of the column pair $(j_1, j_2)$ is defined by the column entropy of $h_{j_1}^c(\cdot) + \max(h_{j_1}^c)h_{j_2}^c(\cdot)$. Then we define $H_2$ to be the minimum of such pair-entropies for all $\binom{3}{2}$ pairs of columns.

Pairwise entropy can be trivially generalized to triple-wise and so on. If we have $c$ columns we may compute each of $H_1, \ldots, H_c$. In practice $H_1$ and $H_2$ may contain all the information we need.

### G.1 WHAT ENTROPIES ARE EXPECTED?

The maximum value for $H_1$ is $\log k$, in the case of a uniform distribution over clusters. The maximum value for $H_2$ is $\log \binom{k}{2} \approx 2 \log k$. (Note $\log n$ is also an upper bound, where $n$ is the number of points in the dataset / rows in the table.)

With the QR method we expect all the entropies to be near their maximum. However, for the Circular Clustering method this is not the case! That would mean we haven't been able to extract any useful cluster information from the data.

Instead we expect entropies close to what one gets from performing Product Quantization (PQ) on a complete dataset. In short:

1. Too high entropy: We are just doing QR more slowly.
2. Too low entropy: We have a table collapse.
3. Golden midpoint: Whatever entropy normal PQ gets.

