# OpenReview forum: "Clustering Embedding Tables, Without First Learning Them"
_ICLR.cc/2023/Conference — Submitted to ICLR 2023_

### Official Review · Reviewer_ZrK7 · 2022-10-24

**Confidence:** 3
**Correctness:** 3
**Technical Novelty And Significance:** 2
**Empirical Novelty And Significance:** 3
**Recommendation:** 5

**Clarity, Quality, Novelty And Reproducibility:**

Clarity can be improved as per my remarks above.
In terms of reproducibility, use of public packages and datasets is very helpful.


**Strength And Weaknesses:**

Strengths:

The background and related work parts are very well written. It is easy to understand the different approaches, even a bit easier than the proposed approach itself.

The method is supported by experiments made on two publicly available datasets.

Figure 5 with accuracy against the number of parameters serves as complexity analysis of the cost of storage.

I liked the authors also explain what they tried and did not work.

Weaknesses:

I think the clarity of the paper can be improved for the part describing the model itself:

Figure 1 is supposed to describe the proposed model “Clustered Compositional Embeddings” (CQR) but in the caption the model is denoted as Cluster QR. Two different nominations can be misleading.

Figure 1 and Figure 4 (b) which refer both to CQR do not seem to show the same model.

In Figure 6, “CQR half” is mentioned before introduction (definition is done only in Appendix).

There are several references to training time but it is unclear to me whether this is the one to obtain the embeddings or the one for training the recommender system.

What would be missing is a complexity analysis of the computation time? Figure 6 shows only the impact of 1 ou 2 epochs on the accuracy (depending on the number of model parameters).

I have several questions:

Q1: p.6: How many samples are considered for step 2? 256 * k but how is k defined?
1000 cluster points: is 1000 the size of S? Are the cluster points the centroids so k = 1000? A diagram flow of the 3 steps would be helpful to understand.

Q2: Where does fit the analysis (section 4) compared to the 3 steps described before? It seems the result of the clustering is used for sparse approximation of $T_i$ but the authors say later: “it could also be done in other ways”. And theorem 1 which follows brings guarantees for a random $H_i$.

Q3: p.7: Figure 5 is with which Criteo dataset? Two are introduced.
“Our method was able to reach the baseline”: what is the baseline? We have to wait for paragraph 5.2 for a description of the baseline. Can we have a reference in the caption for clarity? Or indicate that dashed points are for the baseline?

Q4: p.8: Dataset paragraph: How the pre-hashing is made for Terabyte dataset?

Q5: Do you have reference(s) on the fact recommendation systems are not trained until convergence?

Minor (typos):
-p.3, fig.2: missing bracket for reference to the DLRM model \citep instead of \cite?
-p.9, fig 6 (b): missing “s” to “2 epoch”.


**Summary Of The Paper:**

This paper proposes a new quantization method to deal with high dimensional categorical features in recommendation systems.


**Summary Of The Review:**

I missed several key points of the paper:
1) how CQR works overall from the paper. I think the authors can clarify because the related work section was very well explained.
2) The link of theorem 1 for a random matrix instead of à learned one with CQR

---

> ### Author Response · Authors · 2022-11-19
> **Revision 1**
>
> Thank you for the kind words on our method and results.
>
> We have made a major revision of the paper, rewriting multiple sections and redone the main Figure 1.
>
> > Figure 1 is supposed to describe the proposed model “Clustered Compositional Embeddings” (CQR) but in the caption the model is denoted as Cluster QR. Two different nominations can be misleading.
>
> Thank you. We have fixed this issue now along with what is hopefully a general improvement in consistency.
>
> > Figure 1 and Figure 4 (b) which refer both to CQR do not seem to show the same model.
>
> Figure 4b is the method we call “QR Hybrid” which is a step towards CQR. Hopefully this is more clear now that we have replaced Figure 1.
>
> > In Figure 6, “CQR half” is mentioned before introduction (definition is done only in Appendix).
>
> Thank you. This is now fixed.
>
> > There are several references to training time but it is unclear to me whether this is the one to obtain the embeddings or the one for training the recommender system.
>
> We always train the recommender system end-to-end. Hopefully this is more clear in the updated paper.
>
> > What would be missing is a complexity analysis of the computation time? Figure 6 shows only the impact of 1 ou 2 epochs on the accuracy (depending on the number of model parameters).
>
> We find that a classical “complexity analysis” (like the one done in the DHE paper) is quite misleading, since the performance of embedding table algorithms is entirely down to memory speed. That is, as we reduce the number of parameters, we can use fewer GPUs which reduce the communication overhead. That is the main “complexity” analysis we are interested in.
>
> In terms of pure computation, the QR based methods are very similar and in our experience only made a 5-10% difference when trained together with the rest of the recommendation system. Only the DHE method was about 4 times slower.
>
> > Q1: p.6: How many samples are considered for step 2? 256 * k but how is k defined? 1000 cluster points: is 1000 the size of S? Are the cluster points the centroids so k = 1000?
>
> We consider 256 * 1000 many samples for finding k=1000 centroids. The set S is the set of the 256 * 1000 samples we consider. In the original draft, we were using “cluster points” to mean centroids, which is confusing. We now only use the term centroids.
>
> > A diagram flow of the 3 steps would be helpful to understand.
>
> This is now in the updated Figure 1.
>
> > Q2: Where does fit the analysis (section 4) compared to the 3 steps described before? It seems the result of the clustering is used for sparse approximation T_i of  but the authors say later: “it could also be done in other ways”. And theorem 1 which follows brings guarantees for a random H_i.
>
> We have rewritten section 3 and 4 to better explain the role of clustering.
>
> > Q3: p.7: Figure 5 is with which Criteo dataset? Two are introduced.
>
> Figure 5 is the Criteo’s Kaggle dataset. We now have replaced the original “Criteo dataset” to “Kaggle dataset”.
>
> >“Our method was able to reach the baseline”: what is the baseline? We have to wait for paragraph 5.2 for a description of the baseline. Can we have a reference in the caption for clarity? Or indicate that dashed points are for the baseline?
>
> We added a reference to section 5.2, the description of the baseline.
> We also changed the name of the dashed line from “Full” to “Baseline”.
>
> > Q4: p.8: Dataset paragraph: How the pre-hashing is made for Terabyte dataset?
>
> It is just done by modding by 10M, as implemented in the original DLRM code.
>
> > Q5: Do you have reference(s) on the fact recommendation systems are not trained until convergence?
>
> What we mean to say is that they are only trained for one epoch. This follows DLRM (Naumov et al. 2019) and (Shi et al. 2019). It is easy to see (training DLRM) that overfitting sets in immediately after the second epoch starts.
>
> > Minor (typos): -p.3, fig.2: missing bracket for reference to the DLRM model \citep instead of \cite? -p.9, fig 6 (b): missing “s” to “2 epoch”.
>
> Thank you. This is now fixed.
>
> > how CQR works overall from the paper. I think the authors can clarify because the related work section was very well explained.
>
> Hopefully the new Figure 1 helps, as well as the updated Sections 3 and 4.
>
> > The link of theorem 1 for a random matrix instead of à learned one with CQR
>
> Section 4 now contains a lot more on this.

---

> > ### Comment · Reviewer_ZrK7 · 2022-12-02
> > **Thank you very much to the authors for their answers to our reviews and for improving the paper during the rebuttal period**
> >
> > Thank you very much to the authors for their answers to our reviews and for improving the paper during the rebuttal period. The modifications bring valuable content and clarification.
> > I read also carefully the other reviews and the corresponding answers. However, I keep my recommendation "marginally below the acceptance threshold" as long as there is no explicit feedback from other reviewers that their concerns are addressed (my understanding for now is that it is not 100% addressed).

---

### Official Review · Reviewer_ZdXj · 2022-10-29

**Confidence:** 3
**Correctness:** 3
**Technical Novelty And Significance:** 2
**Empirical Novelty And Significance:** 2
**Recommendation:** 5

**Clarity, Quality, Novelty And Reproducibility:**

Clarity and Quality: Paper is poorly written.
Novelty: Even though there are no groundbreaking ideas, paper uses existing ideas in novel way.

**Strength And Weaknesses:**

Strengths:
1) Authors give thorough literature survey and give intuition on how they proposed their method
2) Authors have strong empirical evidence on datasets used

Weaknesses:
1) Paper is poorly written. Each of the sections separately can be well written but it is hard to get a connection between each section.
2) Some of the details are missing to properly evaluate the paper
3) It is not clear if some of the claims are supported by empirical evidence (e.g. least square setting) and it is also not clear if the evaluation metrics used are the right ones for the imbalanced dataset (e.g. BCE).


**Summary Of The Paper:**

Authors point out the pros and cons of two existing methods for table compression. Authors demonstrate that combining hashing and clustering based algorithms provides the best of both worlds. Authors prove that this technique works rigorously in the least-square setting.

**Summary Of The Review:**


1) "Unfortunately, in our experiments, DHE did not perform as well as the other methods": Have authors tried other deep hashing methods [1-4]?

2) There is a big jump from section 3 to section 4. What is X and what is Y in the context of hashing or clustering embedding tables that authors talked about in the section 3? It is left to the reader to make the connection.

3) "We thus give results both for 1 epoch training, 2 epoch training and “best of 10 epochs”: Can authors define what's best here? Is it on a validation set or test set?

4) Figure 5 shows the results with respect to the number of parameters. Would it be possible to share actual computation time comparison?

5) How was hyperparameter selection done for each of the methods, if any? For example, learning rate.

6) Is BCE the right metric in this case? As it is a click data, it would be too imbalanced. Can authors share more details here?

7) Most of the plots contain standard deviations. What are these? Were each of the experiments done multiple times?

8) "We prove that this technique works rigorously in the least-square setting": How is BCE loss used in experiments related to this claim?

8) Writing: Each section seems disconnected from other sections. Section 2 gives a good overview of existing literature and sets up the proposed method introduced in section 3. But there is a huge jump from section 3 to 4. Section 4 looks like a completely new paper. Start of section 5 is abrupt too and it is not clear what authors are trying to convey without knowing details of the exact dataset and how evaluations of each of the methods are done. Some of the dataset details are missing and not connected to variables used in section 4 where actual optimization is described. Writing should be improved so that readers can easily understand what authors are trying to convey.

[1] Cao, Yue, Mingsheng Long, Bin Liu, and Jianmin Wang. "Deep cauchy hashing for hamming space retrieval." In Proceedings of the IEEE Conference on Computer Vision and Pattern Recognition, pp. 1229-1237. 2018.
[2]Jang, Young Kyun, Geonmo Gu, Byungsoo Ko, Isaac Kang, and Nam Ik Cho. "Deep Hash Distillation for Image Retrieval." In European Conference on Computer Vision, pp. 354-371. Springer, Cham, 2022.
[3] Boyles, Levi, Aniket Anand Deshmukh, Urun Dogan, Rajesh Koduru, Charles Denis, and Eren Manavoglu. "Semantic Hashing with Locality Sensitive Embeddings." (2020).
[4] Hoe, Jiun Tian, Kam Woh Ng, Tianyu Zhang, Chee Seng Chan, Yi-Zhe Song, and Tao Xiang. "One loss for all: Deep hashing with a single cosine similarity based learning objective." Advances in Neural Information Processing Systems 34 (2021): 24286-24298.

---

> ### Author Response · Authors · 2022-11-19
> **Revision 1**
>
> Thank you for the kind words on our method and results.
>
> We have made a major revision of the paper, making the sections more fluent and detailed.
>
> > It is not clear if some of the claims are supported by empirical evidence (e.g. least square setting)
>
> Our main claim is empirical: We test our method on real world datasets using real world recommendation models.
> To improve the theoretical understanding of our method, we also make an analytical claim in the least squares setting.
> In the revision we have added some experiments in the least squares setting as well (Figure 5), but these are less of a claim and more a guide of intuition.
>
> > it is also not clear if the evaluation metrics used are the right ones for the imbalanced dataset (e.g. BCE). Is BCE the right metric in this case? As it is a click data, it would be too imbalanced. Can authors share more details here?
>
> In the literature both BCE and AUC are common measures. However, we have yet to see a paper in which the choice of one over the other makes any significant difference.
> We have rerun our experiments with AUC and added the graphs to the appendix for comparison.
>
> > "Unfortunately, in our experiments, DHE did not perform as well as the other methods": Have authors tried other deep hashing methods [1-4]?
>
> We thank the reviewer for the interesting papers [1-4].
> Unfortunately these papers all seem to focus on the subtly different problem of learning a (locality sensitive) hash for faster image retrieval.
> With some variation they all propose a framework along the lines of:
> Embedding the images with a CNN
> Learn a function from the dense NN representation to a hash code
> Use these hash codes for image retrieval.
>
> Such a learned hash function would indeed make our task of embedding discrete data for recommendation systems easier, since we could use the same embedding for IDs that hash together.
> For our data (discrete IDs from a large universe) there is no simple way to perform step 1 of embedding the data into a NN. Indeed getting the data into a NN is _the main problem we are solving_. Once our method has been applied, and a reasonable dense representation has been found, one may use the methods of papers [1-4] to learn a hash function for fast retrieval.
>
> > Figure 5 shows the results with respect to the number of parameters. Would it be possible to share actual computation time comparison?
>
> Reducing the number of parameters is the main bottleneck we are trying to solve. The inference time of the algorithms is very hardware specific, and things like the number of GPUs used to shard the tables makes a huge difference.
>
> For what it’s worth, in our experiments the QR based methods were all very similar (within 10%) in terms of inference time. Only DHE was about 4 times slower.
>
> > "We thus give results both for 1 epoch training, 2 epoch training and “best of 10 epochs”: Can authors define what's best here? Is it on a validation set or test set?
>
> For “best of 10 epochs”, we train our model for 10 epochs, and throughout the training, we evaluate the model on the test set for 60 times, and pick the best performance as measured in BCE.
>
> > How was hyperparameter selection done for each of the methods, if any? For example, learning rate.
>
> We use the choice for default hyperparameters used in DLRM as well as papers that build upon it. This means the learning rate is 0.1 and the optimizer is vanilla SGD.
>
> > Most of the plots contain standard deviations. What are these? Were each of the experiments done multiple times?
>
> Each experiment is done three times. The bands shown in the plots are max and min, while the middle line is the average.
>
> > "We prove that this technique works rigorously in the least-square setting": How is BCE loss used in experiments related to this claim?
>
> We have completely rewritten the analysis section, and hopefully things are better now.
> The answer is that we imagine there is some “optimal” embedding table that gives the smallest BCE loss on the task at hand. In the theoretical section we show that our algorithm is able to perfectly reconstruct such a table when a linear relationship exists between the data and the intended (but unknown) table. The use of least-squares is less important, as it is simply a vehicle to show convergence, and the final 1+epsilon convergence in least-squares translates to an O(epsilon) approximation to whatever downstream loss, as long as the network is Lipshitz.

---

> > ### Comment · Reviewer_ZdXj · 2022-12-07
> > **Response to authors**
> >
> > Some of my concerns on the performance metric BCE/AUC, related hashing methods [1-4], hyperparameters, plots containing standard deviations have been addressed. But the paper still lacks clarity, confusion/gap between theory and experiments. I am improving my score from 3 to 5.

---

### Official Review · Reviewer_7zzb · 2022-11-08

**Confidence:** 4
**Correctness:** 3
**Technical Novelty And Significance:** 2
**Empirical Novelty And Significance:** 2
**Recommendation:** 5

**Clarity, Quality, Novelty And Reproducibility:**

The paper is mostly clear. The proposed method is novel and can be easily implemented.

**Strength And Weaknesses:**

**Strength**
1. The authors propose a simple method to bridge the gap between post-training quantization and hashing-based methods for embedding compression.
2. Existing methods are fully surveyed and introduced in the paper.

**Weaknesses**
1. The empirical comparisons are incomplete.
(a) The Tensor Train method (Yin et al., 2021) is not compared in the experiments.
(b) The results of PQ is missing in Figure 5 and Figure 6(b).
2. The step-1 of CQR still requires firstly training the embedding tables, and hence the current paper title and the statement, "Our algorithm is the first that deviates from random (or fixed) sketching as the first step of embedding", are both imprecise.
3. Figure 1 is confusing and cannot help understanding the proposed method.
4. Why are the results of PQ in Figure 6(a) and Figure 7(a) so different? It seems that PQ was not correctly applied to the Terabyte dataset.

**Minor problem**

"increases" -> "decreases", end of page 6.

**Summary Of The Paper:**

The paper proposed a new method called Clustered Compositional Embeddings (CQR) for learning compressed embedding tables. The method is a QR concat method with a set of specially initialized embedding tables and carefully chosen hash functions. The authors claim that CQR may achieve compression ratios close to those of post-training quantization with the training time memory reductions of hashing-based methods.

**Summary Of The Review:**

The proposed method may be useful in practice. However, I wish the authors can address the issues mentioned above.

---

> ### Author Response · Authors · 2022-11-19
> **Revision 1**
>
> Thank you for the kind words on our method and results.
>
> We have made a major revision of the paper, making the sections more integrated and with more clarity about what's empirical and what's theoretical.
>
> > The empirical comparisons are incomplete. (a) The Tensor Train method (Yin et al., 2021) is not compared in the experiments.
>
> In their paper, Yin et al claim only 112x compression on Criteo, which is substantially less than the >1000x compression claimed in ROPE and shown by our methods in Figure 6.
>
> > (b) The results of PQ is missing in Figure 5 and Figure 6(b).
>
> We decided it was most fair to Product Quantization only to apply it when the other methods were tested for a single epoch. This is because PQ is just clustering on the “Full table” baseline, which is only trained for one epoch. (Because it overfits if trained any longer.)
> We provide Figure 7a and 7b side by side to make it easy to compare PQ with our hashing and clustering based methods.
>
> > The step-1 of CQR still requires firstly training the embedding tables, and hence the current paper title and the statement, "Our algorithm is the first that deviates from random (or fixed) sketching as the first step of embedding", are both imprecise.
>
> This is a good point. What we mean to say is that we don’t _fully_ train the tables before we cluster them. This is what gives our algorithm a head up against PQ, in that the interleaved training and clustering ultimately allows finding a much better clustering.
>
> We would like to change our paper title to something like “Clustering Embedding Tables, Without First _Fully_ Learning Them.” But since we can’t update the title at this point, we have opted to improve the sentence you mention from our related work.
>
> > Figure 1 is confusing and cannot help understanding the proposed method.
>
> We agree the figure was confusing and have now completely redone it.
>
>
> > Why are the results of PQ in Figure 6(a) and Figure 7(a) so different? It seems that PQ was not correctly applied to the Terabyte dataset.
>
> We too are somewhat puzzled by this. The AUC metric showed the same pattern. We added this now in Appendix F.
> We think the Terabyte dataset may simply be too hard for a single training + clustering iteration (as done by PQ) to learn. Comparing with our (new) Figure 5b, we see that it sometimes takes many training + clustering iterations to improve measurably over simple hashing.

---

### Author Response · Authors · 2022-11-19
**Common concerns**

We take it that the reviewers like the simplicity of our algorithm and the writing of our background sections.
Meanwhile the writing of section 3 and 4 was not good enough, and Figure 1 was confusing.
We have rewritten the sections and redone the figure.

The reviewers also commented positively on the strong empirical evidence on the publicly available datasets.
We hope they will similarly enjoy the improved proofs and clarity of the theoretical section.

---

### Decision · Program_Chairs · 2023-01-20

**Decision:**

Reject

**Justification For Why Not Higher Score:**

NA

**Justification For Why Not Lower Score:**

NA

**Metareview: Summary, Strengths And Weaknesses:**

The paper presented Clustered Compositional Embeddings (CQR) compressing embedding tables. The method combines QR concat  with hashing based methods to achieve the best of both worlds. The paper argues that CQR may achieve compression ratios close to those of post-training quantization with the training time memory reductions of hashing-based methods.

However, the paper during submission was not written properly and reviewers cannot verify a lot of interesting claims and arguments. Their reviews clearly points out their lack of excitement.  The authors did a major revision of the paper post response.


**Summary Of Ac-Reviewer Meeting:**

Unfortunately, a major revision is like another submission which needs a separate review process to verify the claims, settings, and novelty.

We hope the reviewers benefited from the process and will help them improve the paper for future submissions.